# Synthesis and Properties of (1*R*(*S*),5*R*(*S*),7*R*(*S*),8*R*(*S*))-1,8-Bis(hydroxymethyl)-6-azadispiro[4.1.4.2]tridecane-6-oxyl: Reduction-Resistant Spin Labels with High Spin Relaxation Times

**DOI:** 10.3390/ijms241411498

**Published:** 2023-07-15

**Authors:** Yulia V. Khoroshunova, Denis A. Morozov, Danil A. Kuznetsov, Tatyana V. Rybalova, Yurii I. Glazachev, Elena G. Bagryanskaya, Igor A. Kirilyuk

**Affiliations:** 1N.N. Vorozhtsov Institute of Organic Chemistry SB RAS, Academician Lavrentiev Ave. 9, 630090 Novosibirsk, Russia; horoshunova@nioch.nsc.ru (Y.V.K.);; 2Department of Physics, Novosibirsk State University, Pirogova Str. 1, 630090 Novosibirsk, Russia; 3Voevodsky Institute of Chemical Kinetics and Combustion SB RAS, Institutskaya 3, 630090 Novosibirsk, Russia

**Keywords:** dispiro[pyrrolidine-2,1′-cyclopentane-5,1″-cyclopentane], 6-azadispiro[4.1.4.2]tridecane, nitrones, 1,3-dipolar cycloaddition, pyrrolidine nitroxides, sterically shielded nitroxides, spin labels

## Abstract

Site-directed spin labeling followed by investigation using Electron Paramagnetic Resonance spectroscopy is a rapidly expanding powerful biophysical technique to study structure, local dynamics and functions of biomolecules using pulsed EPR techniques and nitroxides are the most widely used spin labels. Modern trends of this method include measurements directly inside a living cell, as well as measurements without deep freezing (below 70 K), which provide information that is more consistent with the behavior of the molecules under study in natural conditions. Such studies require nitroxides, which are resistant to the action of biogenic reductants and have high spin relaxation (dephasing) times, T_m_. (1*R*(*S*),5*R*(*S*),7*R*(*S*),8*R*(*S*))-1,8-bis(hydroxymethyl)-6-azadispiro[4.1.4.2]tridecane-6-oxyl is a unique nitroxide that combines these features. We have developed a convenient method for the synthesis of this radical and studied the ways of its functionalization. Promising spin labels have been obtained, the parameters of their spin relaxation T_1_ and T_m_ have been measured, and the kinetics of reduction with ascorbate have been studied.

## 1. Introduction

Site-directed spin labeling coupled to Electron Paramagnetic Resonance spectroscopy (SDSL–EPR) is a rapidly expanding powerful biophysical technique to study biomolecules in physiologically relevant environments [1,2,3,4,5,6]. These methods are based on site-specific introductions of paramagnetic labels (unpaired electrons) into biomolecules of interest with subsequent investigation using various EPR techniques. The latter give inter-spin distances, solvent accessibility, the polarity of its immediate environment, and the dynamics of the labeled region, which are complimentary to information obtained by other methods of structural biology, such as NMR, X-ray crystallography, and cryo-electron microscopy. This helps to identify biologically important conformations of large biomolecules, especially membrane and intrinsically disordered proteins. Nitroxides are the most widely used spin labels [7]. Unlike other spin labels, they can be used both to study local dynamics in biomolecules at physiological temperatures and to measure inter-spin distances [5]. Examples of specific applications of nitroxide spin labels to structural biology studies of protein can be found in hundreds of publications [2].

The best way to obtain correct information about the native structure and functions of biomolecules implies their study in natural conditions—in a living cell. Currently, a set of evidence has been accumulated for the difference in protein structure and function in model and natural conditions [8,9,10]. For this reason, SDSL–EPR experiments in living cells attract more and more attention. A serious obstacle to the use of nitroxide spin labels in living cells is the rapid reduction of the nitroxide into diamagnetic compounds by low molecular weight reductants and enzymatic systems, typically found within cells [11]. Resistance of nitroxides to bioreduction can be strongly improved via introduction of several bulky alkyl (larger than methyl) substituents to the α-carbon atom of the nitroxide group [12]. A number of reduction-resistant spin labels for in-cell application have been developed based on tetraethyl nitroxides [13,14,15,16,17,18,19].

Several EPR methods have been developed for inter-spin distance measurement [20]. All of them are dependent on electron spin relaxation parameters of spin labels [2,21,22]. In Pulsed Electron–Electron Double Resonance (PELDOR or DEER) technique, which is currently the most popular approach for distance measurements, the maximal distance one can measure and the precision of the distance distribution are determined by the phase memory time (T_m_) of the spin label [22]. The T_m_ parameter is temperature-dependent and must be as high as possible. To achieve optimal performance using conventional tetramethyl or tetraethyl spin labels, the measurements must be carried out at 40–65 K, because higher temperature rotation of the alkyl groups leads to a decrease in T_m_. This rotation is impossible in nitroxides with spirocyclic moieties at α-carbons of nitroxide group, and these spirocyclic spin labels can be used for measurements at much higher temperatures (125 K) and even at room temperature [23]. Regretfully, the spirocyclic spin labels demonstrate much lower resistance to bioreduction compared to tetraethyl nitroxides [24].

According to expert estimates, labels that can eliminate the need for data acquisition at cryogenic temperatures and labels that can enter and survive in the cellular environment are of current and future interest [2]. The attempts to improve reduction resistance of spirocyclic nitroxides have been a subject of recent research [25]. The literature data on the rate constants of chemical reduction of representative dispirocyclic nitroxides and tetraethyl nitroxides are listed in the Figure 1. The data demonstrate that reduction resistance of the nitroxide **1** by far exceeds those of other spirocyclic nitroxides. The nitroxide group in **1** is stabilized with two spiro-(2-hydroxymethyl)cyclopentane moieties with the hydroxymethyl groups directed towards N-O^•^. In addition, this nitroxide showed the highest longitudinal relaxation time T_1_ among a broad set of various nitroxides [26]. This parameter is very important for distance measurement using Saturation Recovery (SR) method, another pulsed EPR technique widely used in structural biology [2,21].

The nitroxide **1** was prepared from 3,4-di-*tert*-butoxypyrroline 1-oxide [31] via repetitive sequence of procedures: pent-4-en-1-ylmagnesium bromide addition, intramolecular 1,3-dipolar cycloaddition, isoxazolidine ring opening and oxidation [27,30]. The literature protocol implies multi-step synthesis with low overall yield (ca. 5%) from commercially available L-tartaric acid, making **1** unfavorable object for further chemical transformations [27,30].

Recent advances in condensation, cyclization, and dipolar cycloaddition cascade chemistry [32] allowed us to develop convenient and scalable protocol for synthesis of dispirocyclic nitroxide **5**, a 3,4-unsubstituted analog of **1**, with overall yield 43% from commercially available 4-chlorobutyryl chloride (**6**). Chemical properties of the nitroxide **5** were studied and several dispirocyclic spin labels were prepared. The new nitroxides showed improved resistance to reduction compared to **1**. The spin relaxation times T_1_ and T_m_ were measured for some of the nitroxides.

## 2. Results and Discussion

### 2.1. Synthesis

The (1*R*(*S*),8*R*(*S*))-6-oxa-5-azatricyclo[6.3.0.01,5]undecane (**8**) (6a*R*(*S*),9a*R*(*S*))-hexahydro-1*H*,6*H*-cyclopenta[*c*]pyrrolo[1,2-*b*]isoxazole) was prepared in two steps as enantiomeric mixture according to the literature protocol [32] (Figure 1). This allowed us to avoid multi-step synthesis.

Subsequent oxidative isoxazolidine ring opening with m-cloroperoxybenzoic acid (*m*-CPBA) afforded aldonitrone **9**. The reaction was carried out in dry dichlorometane (DCM) using dry reagent to prevent formation of hydroxamic acid. The structure of **9** was confirmed with IR, UV, ^1^H, and ^13^C NMR (see Section 3 and Appendix A) and X-ray diffraction data (Figure 2).

We have previously published a set of procedures to convert 1-pyrroline 1-oxides into pyrrolidine nitroxides with spiro-(2-hydroxymethyl)cyclopentane moiety [30]. Here, we used trimethylsilyl protection of the hydroxy group in order to avoid unproductive consumption of the organometallic reagent (Figure 2). The crude silylated nitrone **10** was treated with 1.5-fold excess of pent-4-en-1-ylmagnesium bromide to give hydroxylamine **11** and the latter was in situ oxidized with air oxygen in presence of Cu^2+^.

The nitrone **12** easily undergoes intramolecular 1,3-dipolar cycloaddition affording single product. The best yield was achieved upon heating to reflux in toluene in presence of 2,2,6,6-tetramethylpiperidine-1-oxyl (TEMPO). Intramolecular cycloaddition in 2-pent-4-en-1-ylpyrroline 1-oxides was reported to give a single regioisomer [27,30,33]. Similarly, the ^1^H and ^13^C NMR spectra and ^1^H-^1^H and ^1^H-^13^C correlations revealed the signals of two different CH-CH_2_O moieties (see Section 3 and Appendix A), confirming formation of (hexahydro-1*H*-spiro[cyclopenta[*c*]pyrrolo[1,2-*b*]isoxazole-3,1′-cyclopentane]-2′-yl) methanol system. However, one could not exclude the possibility of C=C bond approaching the plane of the nitrone group from different sides with the formation of *cis*- and *trans*-isomers relative to the position of the hydroxymethyl group. To assign a structure to the cycloadduct ^1^H-^1^H NOESY NMR spectra were recorded (Appendix A). ^1^H-^1^H NOESY correlation showed a cross-peak between protons of ^6^CH_2_ and ^15^CH_2_ groups, while no cross-peak was observed between ^11^CH_2_ and ^6^CH_2_ protons (Figure 3). Apparently, the hydroxymethyl group prevents C=C from approaching the nitrone group, and the formation of a cycloadduct is possible only through the transition state **15** (Figure 2).

Reductive scission of the isoxazolidine ring N-O bond with Zn-AcOH system afforded **14**, which was isolated as a colorless crystalline solid. Half of the signal set was observed in ^1^H and ^13^C NMR spectra of **14**, indicating symmetrical structure. Single-crystal X-ray diffraction data of this compound showed C_2_-symmetry of the molecule, which confirms our previous assignment of structure to **13**. Thus, despite the absence of bulky substituents at positions 3 and 4 of the pyrroline ring, the intramolecular 1,3-dipolar cycloaddition reaction in **12** proceeds stereospecifically.

Previously, we reported that oxidation of 2,2-disubstituted 1-azaspiro[4.4]nonan-6-ylmethanols to nitroxides with *m*-CPBA may be accompanied with conversion of hydroxymethyl group into aldehyde, and acylation of the hydroxy group(s) before oxidation increases the nitroxide yield [30]. Following this procedure, **14** was heated with acetic anhydride (Ac_2_O) to give **16**, which was then oxidized with *m*-CPBA (Figure 3). The reaction afforded two nitroxides. The main product **17** was isolated as yellow crystals with 86% yield. The structure of **17** was confirmed by single-crystal X-ray diffraction data (Figure 2).

The structure of the minor product **18** was assigned on the basis of ^1^H and ^13^C NMR, and ^1^H-^1^H COSY, ^1^H-^13^C HSQC, and ^1^H-^13^C HMBC spectra were acquired after reduction of the nitroxide to corresponding amine with Zn in presence of trifluoroacetic acid according to the earlier described procedure [18] (see Section 3 and Appendix A) and confirmed with high-resolution mass-spectrum and IR spectral data (Appendix A). We have previously observed the formation of similar dehydrogenated nitroxide upon oxidation of (2,2-dimethyl-1-azaspiro[4.4]nonan-6-yl)methyl acetate, and proposed a mechanism implying proton abstraction from intermediate oxoammonium cation [30].

Nitroxide **17** was then subjected to alkaline hydrolysis to give **5** with nearly quantitative yield. The structure of **5** was confirmed by single-crystal X-ray diffraction data (Figure 4). The overall yield of this nitroxide starting from commercially available 4-chlorobutyryl chloride and 5-bromo-1-pentene exceeds 40%, which makes it an attractive material for the synthesis of spin labels. 

Modification of the hydroxymethyl groups seems to be the simplest way to functional derivatives capable of binding to biomolecules. In our recent study, we demonstrated that activation of the hydroxyl group to nucleophilic substitution in 1-unsubstituted pyrrolidines and corresponding alkoxyamines, acyloxyamines, or nitroxides with spiro-(2-hydroxymethyl)cyclopentane moiety always leads to cyclization, which may be followed by rearrangement [34]. Here, we studied the oxidation of hydroxymethyl groups to carboxylic ones, and the alkylation/acylation of hydroxyl groups.

Attempts to oxidize the hydroxymethyl groups in **5** using TEMPO—Sodium chlorite system [35] were unsuccessful, leading to strong tarring. This may occur due to the formation of an oxoammonium cation, which can cause oxidative transformations in the side chains (cf. [30] and above). Therefore, we used the diamagnetic precursor **14** to prepare the desired nitroxide with carboxylate groups. Direct oxidation of **14** with Jones reagent afforded crude **20** as a colorless viscous oil (Figure 4). The NMR spectra showed half of the signal set, showing that the compound remained a racemic mixture (no other diastereomers formed). The pure crystalline sample was isolated via column chromatography and crystallization from methanol-ethyl acetate mixture 50:1, and the structure was confirmed by single-crystal X-ray diffraction data (Figure 4).

Attempts to oxidize of **20** either with *m*-CPBA or H_2_O_2_/Na_2_WO_4_ were not successful, so the crude **20** was dissolved in methanol saturated with HCl and diester **21** was isolated. Oxidation of **21** with *m*-CPBA afforded **22**, which was isolated as a yellow crystalline solid. The ^1^H NMR spectrum acquired after reduction of freshly prepared nitroxide with Zn/CF_3_COOH showed half of the signal set with significant downfield shift (due to protonation) as compared to spectrum of **21**. The structure of **22** was confirmed by single-crystal X-ray diffraction data (Figure 4). Alkaline hydrolysis of **22** gave an inseparable mixture of structurally related compounds with total yield 42% (Figure 5). The element analysis of the mixture corresponded to the formula C_14_H_20_NO_5_, which allowed us to assume that the product was a mixture of isomers. The ^13^C NMR spectrum of the mixture after reduction with Zn/CF_3_COOH corresponded to a mixture of three isomers, two symmetric and one asymmetric dicarboxylic acids (Appendix A). This picture corresponds to a mixture of three isomers, two symmetric, and one asymmetric dicarboxylic acids. The ratio of the isomers was estimated using integrals of the signals of methine hydrogens at 3.02–3.30 ppm in ^1^H NMR spectrum.

Inversion of the asymmetric center adjacent to the ester group may result from C-H acidity. TLC analysis of samples of **22** after long-term storage showed the emergence of two compounds with close R_f_, probably the isomers. The formation of these compounds accelerates in the presence of a base or LiI; however, these reactions were accompanied by tarring. Presumably, in alkaline solution, isomerization proceeds faster than hydrolysis, resulting in nearly statistical ratio of isomers.

It was shown that nitroxide group can decrease pK_a_ of the acidic center in two σ-bond distance by 2.5 orders of magnitude compared to corresponding methoxyamine derivative [36]. Thus, reduction of the nitroxide group may slow down the isomerization. To verify this hypothesis, the freshly prepared nitroxide **22** was reduced to corresponding hydroxylamine with ascorbic acid in oxygen-free conditions, and then, potassium hydroxide was added (Figure 6). After the hydrolysis was complete, the products were oxidized with air oxygen, acidified, and extracted. Dicarboxylic acid **23a** was a major component of the resulting mixture and it was isolated with the yield 45%. The structure of **23a** was confirmed by single-crystal X-ray diffraction data (Figure 5).

Inversion of the asymmetric center at the ester group in **22** leads to a loss of configuration that provides higher hindrance to the nitroxide group. Despite succeeding in isolating pure **23a**, it is obvious that activated esters capable of binding to biomolecules, which could be prepared from **23a**, will behave similarly to **22**. Thus, the nitroxides with spiro-(2-carboxy)cyclopentane moieties are not optimal for spin labeling.

Exploring the possible ways to functional derivatives of **5**, we tried several alkylation and acylation reactions. A reaction of nitroxide alcohols with carbonyldiimidazole (CDI) was used for binding to primary amino groups [37,38]. A reaction of **5** with CDI afforded the nitroxide **26** with an excellent yield (Figure 7). The structure of **26** was confirmed by single-crystal X-ray diffraction data (Figure 5). A reaction of **26** with *N*,*N*-dimethyl-1,3-diaminopropane gave **27** with 55% yield.

Another carbamate derivative **28** was prepared via treatment of **5** with methyl 3-isocyanatopropionate with quantitative yield. After alkaline hydrolysis of the ester groups corresponding dicarboxylic acid **29** was isolated. The structure of the nitroxides **27**, **28**, and **29** were confirmed by element analyses, and IR spectra data and ^1^H and ^13^C NMR spectra were acquired after reduction with Zn/CF_3_COOH, which showed half of the signal set (See Supporting Information).

Recently, copper-catalyzed azide-alkyne cycloaddition (CuAAC) was successfully used for the site-directed spin labeling of the protein in vivo [39]. The spin labels capable of binding azides were prepared via alkylation of **5** with propargyl bromide (Figure 8). Two nitroxides **30** and **31** were isolated from the reaction mixture. The proposed structures were confirmed with IR spectra, HRMS, and element analysis data. The ability of the spin label **31** to bind to azide-containing biomolecules was demonstrated by the reaction with 2,3,4,6-tetra-*O*-acetyl-β-d-galactopyranosyl azide in analogy to the literature protocols [40,41,42]. The reaction of enantiomerically pure galactose derivative with racemic mixture **31** expectedly gave a mixture of diastereomers **32**, which was not separated. The structure of **32** was confirmed by element analysis, and IR spectral data and ^1^H and ^13^C NMR spectra were acquired after reduction with Zn/CF_3_COOH (See SI). The NMR spectra of two diastereomers are very close: only few signals of carbon atoms of the spirocyclic system differ and it is impossible to assign them to a specific diastereomer. Ammonolysis of **32**, in analogy with the literature procedure [42], afforded spin-labelled galactose **33**, which was isolated as a glassy solid. To acquire ^1^H and ^13^C NMR spectra, the nitroxide sample was reduced with Zn in presence of formic and oxalic acids mixture. Similarly to the above-mentioned for **32**, most of the signals of the two diastereomers were not resolved, and very few slightly differed in chemical shift (see Section 3 and Appendix A).

### 2.2. CW EPR Spectra

CW EPR spectra of nitroxides **5** and **33** are shown in Figure 6 and their parameters obtained from simulations of experimental spectra using EasySpin software are given in Table 1. For comparison spectra of 3-carboxy-Proxyl (**34**) and 2,2,5,5-tetramethyl-3-(((1-((2*R*,3*R*,4*S*,5*R*,6*R*)-3,4,5-trihydroxy-6-(hydroxymethyl)tetrahydro-2*H*-pyran-2-yl)-1*H*-1,2,3-triazol-4-yl)methoxy)methyl)pyrrolidine-1-oxyl (**35**) (Table 1) (see Section 3 for synthesis and Appendix A and S63 for spectra). Measurements were carried out in aqueous solutions, and in addition for **5** and **34**, the spectra were recorded in toluene (**33** and **35** are poorly soluble in toluene, partition coefficients octanol/water (K_p_) for **5**, **29**, and **33** were 11, 6, and 0.1, respectively (see Section 3 and Appendix A for details). Broadening of the spectral lines is a typical feature of spirocyclic nitroxides (cf. [28,43]). This broadening is caused by the contribution of hyperfine interactions of an unpaired electron with hydrogen nuclei on substituents. In the spectra of **34** and **35** at room temperature, the rotation of methyl groups leads to averaging of hyperfine interactions.

### 2.3. Electron Spin Relaxation Time

Electron spin relaxation times were measured for four radicals **5**, **33**, **34**, and **35** at two temperatures of 80 K and 120 K in water−glycerol solution (1:1); the data are shown on Figure 7 and are listed in Table 2. In Figure 7a, the non-exponential decay of spin echo for both radicals is clearly visible. A similar effect was described by Eaton and coworkers [22,24,26] for the phase memory time of the spin echo of nitroxyl radicals with spirocyclohexyl moieties—the positions 2 and 6 of the piperidine ring. It was shown that the observed spin echo decay curves in a water–glycerole solution (1:1) are described by the following formula: I = I_0_ × exp(−(t/T_2_)^n^, where n > 1, and are determined by the nuclear spin diffusion of solvent protons [22]. This effect has been studied in detail by G. Jeschke et al. in recent years [44,45]. An increase in the contribution to the spin−spin relaxation mechanism of various dynamic processes, such as the rotation of methyl groups, leads to a decrease in the value of n to 1. As expected, spirocyclic nitroxides **5** and **33** showed a less pronounced dependence of relaxation time T_m_ on temperature than radicals with methyl substituents **34** and **35**. An increase in temperature from 80 to 120 K did not lead to a change in the time T_m_ for **33** within the measurement accuracy, while for radical **35** it decreased more than 3-fold from 3.4 to 1.1 μs. This effect is caused by the manifestation of an additional mechanism of electron spin relaxation due to faster rotation of methyl groups with increasing temperature. It can be seen from Figure 7b that, already at 120 K, an exponential decay of spin echo signal is observed for **35** (n ≈ 2.1), whereas for **33** there is no additional contribution to relaxation and n ≈ 2 is retained due to the absence of methyl substituents.

Curiously, binding to galactose does not cause large changes in T_1_, but in all cases leads to a significant increase in T_m_. This effect may be associated with another mechanism of electron spin relaxation due to modulation of *hfi* and g-tensor anisotropy by libration motion of nitroxide molecule [46,47]. The influence of the solvent matrix to the libration motion of nitroxide ring was discussed previously and it was shown that the attachment of long chains to nitroxide ring can affect the libration motion.

### 2.4. Reduction Rate Constants

The kinetics of reduction of nitroxides **5**, **29**, and **33** with ascorbate was studied. The reaction was carried out under argon using large excess of ascorbate in presence of glutathione to suppress possible reverse processes [48] and followed by EPR. The second-order rate constants were calculated from exponential decay of the nitroxide signal (see Section 3 and Appendix A). The resulting values (k_red_, M^−1^s^−1^) for **5**, **29**, and **33** were as follows: (3.2 ± 0.2) × 10^−3^, (1.40 ± 0.06) × 10^−3^ and (2.70 ± 0.06) × 10^−3^, respectively. All the new radicals demonstrate higher resistance to reduction as compared to **1** (Figure 1). Interestingly, removal of electron-withdrawing *tert*-butoxy groups produces only minor effect on the reduction rate (cf. k_red_ for **1** and **5**). Presumably, the electron effect of *t*-BuO-groups is compensated by their influence on the conformation of spirocyclic fragments in analogy to [25]. An increase in steric demand of the substituents adjacent to nitroxide group is known to stabilize the radicals against reduction, enlargement of all neighboring substituents being most efficient [49]. Comparison of the reduction rates for **5**, **33**, and **29** shows similar effect, with **29** being the most resistant to reduction. Presumably, an increase in substituent size in the positions 1 and 8 of the 6-azadispiro[4.1.4.2]tridecane-6-oxyl system stabilize the nitroxide due to proximity of the bulky groups to the radical center. One can expect much higher stabilization effect upon binding of similar nitroxides to large biomolecules.

## 3. Materials and Methods

### 3.1. General

The IR spectra were recorded on a Bruker Vector 22 FT-IR spectrometer (Bruker, Billerica, MA, USA) in KBr pellets (1:150 ratio) or in neat samples (see the Appendix A in this article pp. 6–17, Appendix A) and are reported in wave numbers (cm^−1^). UV spectra were acquired on a HP Agilent 8453 spectrometer (Agilent Technologies, Santa Clara, CA, USA) in ethanol solutions (concentration ~10^−4^) (see the Appendix A in this article pp. 18–21, Appendix A). ^1^H NMR spectra were recorded on a Bruker AV 300 (300.132 MHz), AV 400(400.134 MHz), DRX 500 (500.130 MHz), and Bruker AV 600 (600.300 MHz) spectrometers (Bruker, Billerica, MA, USA). ^13^C NMR spectra were recorded on a Bruker AV 300 (75.467 MHz), AV 400 (100.614 MHz), DRX 500 (125.758 MHz), and Bruker AV 600 (150.945 MHz) spectrometers (see the Appendix A in this article pp. 19–42, Appendix A). All the NMR spectra were acquired for 5–10% solutions in CDCl_3_ or CD_3_OD at 300 K using the signal of the solvent as a standard. NMR spectra of nitroxides for analysis and structure assignment were recorded after reduction with Zn in CD_3_OD-CF_3_COOH (or CD_3_OD-HCOOH-HOOCCOOH mixture) at 65 °C as described in [19] or with Zn and ND_4_Cl in CD_3_OD at 5 °C. Atoms numbering are shown in figures placed on spectra in SI. HRMS analyses were performed using a High-Resolution Mass Spectrometer DFS (Thermo Electron, Waltham, MA, USA).

Reactions were monitored by TLC on precoated TLC sheets ALUGRAM Xtra SIL G/UV_254_ (Macherey-Nagel GmbH & Co. KG, Düren, Germany) using UV light 254 nm, 1% aqueous permanganate, 10% solution of phosphomolybdic acid in ethanol and Dragendorff’s reagent as visualizing agents. Kieselgel 60 (Macherey-Nagel GmbH & Co. KG) or neutral alumina were utilized as an adsorbent for column chromatography.

The X-ray diffraction experiments were carried out on a Bruker KAPPA APEX II diffractometer (graphite-monochromated Mo Kα radiation). Reflection intensities were corrected for absorption by *SADABS2016*/2 program [50] except of **14**, **17**, **20** treated by *SADABS2008*/1 version [50]. The structures were solved by direct methods using the *SHELXS-97* (Sheldrick, 2008) program [51] of **14**, **17**, **20**, and *SHELXT* 2014/5 (Sheldrick, 2014) [52] for the rest ones. All compounds were refined by anisotropic (isotropic for all H atoms) full-matrix least-squares method against *F^2^* of all reflections by *SHELXL2018*/3 [53]. The positions of the hydrogen atoms were calculated geometrically and refined in riding model except of hydrogenes in OH and NH-groups of **14** and **23a** localized from difference map and refined independently with restriction of bond lengths. The presence of two hydrogen atoms on N1 in **20** was proved from the electron density difference map. The asymmetric units of **17**, **22**, and **26** include a half of molecule, the same one of **5** and **14** consists of three and two molecules, respectively. Note that cyclopentane and azalidine cycles in **22** and **23** are statistically disordered due to conformational mobility in approximate ratio 3:1 and 7:1, respectively. The imidazole cycles of **26** are also disordered due to rotational mobility in approximate ratio 1:1. For experimental details see Appendix A.

Crystallographic data for **5**, **9**, **14**, **17**, **20**, **22**, **23a**, and **26** have been deposited at the Cambridge Crystallographic Data Centre as supplementary publication no. CCDC 2269369–2269376. Copy of the data can be obtained, free of charge, on application to CCDC, 12 Union Road, Cambridge CB21EZ, UK (fax: +44-1223336033 or e-mail: deposit@ccdc.cam.ac.uk; internet: www.ccdc.cam.ac.uk).

### 3.2. EPR

CW EPR spectra were recorded at X-band frequencies (~9.4 GHz) on a commercial Bruker spectrometer, Elexsys E 540 (Bruker Corporation, Billerica, MA, USA). Electron spin resonance spectra were recorded with the following settings: frequency, 9.25 GHz; microwave power, 2.0 mW; modulation amplitude, 0.02–0.10 mT; time constant, 40.96 ms; and conversion time, 20.8 ms. Simulations of solution electron spin resonance lines were carried out in the EasySpin software (5.2.35), which is available at http://www.easyspin.org (accessed on 18 May 2023).

Electron spin relaxation times were measured using home-made pulse EPR spectrometers equipped with a flow helium cryostat and temperature control system [54]. T_m_ was measured using a two-pulse electron spin echo (ESE) sequence; T_1_ was measured by an inversion recovery technique with inversion π-pulse and detection of a two-pulse ESE sequence. The π-pulse length was 40 ns.

### 3.3. Kinetic Measurements and Partition Coefficients

For kinetic measurements, stock solutions were prepared in phosphate-citrate-borate buffer (0.5 mM of each): (1) 1 M solution of ascorbic acid and (2) 5 mM solution of glutathione (GSH). Nitroxides were dissolved in the same buffer and diluted to a concentration of 0.2–0.3 mM. The pH was adjusted to 7.5 with NaOH and the solutions were deoxygenated via bubbling with argon. The solutions carefully and quickly mixed in appropriate proportions in a small tube and placed into EPR capillary (50 μL). Oxygen-free conditions were kept permanently. Capillaries were sealed and placed into EPR resonator. EPR experiments were performed on CW EPR X-band spectrometer Bruker ER-200D (9.87 GHz). Spectra were recorded in oxygen-free conditions using the following settings: microwave power 5 mW, modulations amplitude 0.1 or 0.2 mT; time constant 100 ms; conversion time 50.12 ms. The decay of amplitude of low field component of the EPR spectrum was followed in kinetics measurements. Kinetics of decay were fitted with monoexponential function to calculate the first order rate constants. The kinetics measurements were performed at ascorbate concentrations of 100, 200, and 300 mM. The calculated first reaction constants were plotted versus ascorbate concentration; data were fitted with linear dependence. The slope corresponds to second order reaction (See Appendix A). 

For partition coefficients measurements nitroxides were dissolved in 1 mL of distilled water in plastic tube. At this step, the EPR signal was measured as a zero point. Then, the fraction of octanol was added to this solution and shacked for a few minutes to achieve the stationary distribution of radical in the mixture. The tube was shortly centrifuged to separate the octanol and water fractions. The aliquot of radical solution (water phase) was carefully taken with capillary from the bottom of the tube and new EPR spectrum was recorded with the same settings. This procedure was repeated three times for three different octanol contents. The inverse normalized decrease in the EPR signal was plotted versus octanol/water ratio and the slope of the linear fit was used to determine the partition coefficient (see Appendix A).

### 3.4. Synthesis

2,3,4,6-Tetra-*O*-acetyl-β-d-galactopyranosyl azide [40], **8** [32], and 3-hydroxymethyl-2,2,5,5-tetramethylpyrrolidine-1-oxyl (**36**) [55] were prepared according to the literature procedures.

*((1R(S),5R(S),7R(S),8R(S))-1,8-Bis(hydroxymethyl)-6-azadispiro[4.1.4.2]tridecane-6-oxyl* (**5**)**.** Aqueous 10% solution of NaOH (8 mL) was added to a solution of nitroxide **17** (0.335 g, 0.99 mmol) in MeOH (8 mL), and allowed to stand at room temperature until the reaction was complete (up to 12 h). The progress of the reaction was monitored by TLC (silica gel, hexane:Et_2_O = 1:2, visualization with UV-254). The solution was evaporated under reduced pressure. The residue was dissolved in the saturated solution of NaCl (10 mL) and extracted with Et_2_O (4 × 10 mL). The extract was dried with Na_2_SO_4_. After evaporation of the solvent, the crude residue was purified via column chromatography (silica gel, EtOAc) and recrystallized from Et_2_O to give **5** as yellow crystals. Yield: 0.236 g (94%). m.p. 75–77 °C. UV-Vis (λ_max_ (nm)/lg(ε)): 240 (3.24). IR (KBr, cm^−1^): 3276 (O-H). Analysis: found C 65.85, H 9.90, N 5.45; calcd. for C_14_H_24_NO_3_ C 66.11, H 9.51, N 5.51. HRMS (EI/DFS) *m*/*z* [M^+^]: found: 254.1748; calcd. for C_14_H_24_NO_3_: 254.1751.

*[(5R(S),6R(S))-1-Oxido-1-azaspiro[4.4]non-1-en-6-yl]methanol* (**9**). The solution of MCPBA (20.29 g; 117.6 mmol) in dry CH_2_Cl_2_ (200 mL) was added portion-wise within 1 h to the solution of **8** (18.0 g; 117.6 mmol) in dry CH_2_Cl_2_ (100 mL) under argon at −50 °C. The mixture was allowed to warm up to room temperature (TLC control on silica gel, CHCl_3_:CH_3_OH = 10:1, R_f_ = 0.4, visualization with UV-254 and Dragendorff’s reagent). Then the mixture was concentrated in a vacuum, and the residue was separated via column chromatography (Al_2_O_3_, CH_2_Cl_2_) to give **9** as colorless crystals, m.p. 67–68 °C (from Et_2_O). Yield 15.9 g (80%). UV-Vis (λ_max_ (nm)/lg(ε)): 232 (3.91). IR (KBr, cm^−1^): 3222 (O-H), 3074.1 (H-C=), 1593 (C=N). Analysis: found C 63.90, H 8.44, N 8.27; calcd. for C_9_H_14_NO C 63.88, H 8.93, N 8.28. HRMS (EI/DFS) *m*/*z* [M-17] ^+^: found: 152.1069; calcd. for C_9_H_14_NO: 152.1070. ^1^H NMR (400 MHz, CDCl_3_, δ): 1.55 (dddd, J_d1_ = 3.9, J_d2_ = 8.1, J_d3_ = 10.2, J_d4_ = 16.5, 1H ^7^CH_2_); 1.69 (ddd, J_d1_ = 2.8, J_d2_ = 8.1, J_d3_ = 13.1, 1H ^6^CH_2_); 1.78 (dddd, J_d1_ = 3.6, J_d2_ = 3.9, J_d3_ = 8.1, J_d4_ = 12.6, 1H ^8^CH_2_); 1.88 (ddd, J_d1_ = 7.3, J_d2_ = 7.9, J_d3_ = 12.6, 1H ^8^CH_2_); 1.93 (ddddd, J_d1_ = 2.8, J_d2_ = 3.6, J_d3_ = 7.9, J_d4_ = 8.5 J_d5_ = 16.5, 1H ^7^CH_2_); 2.01 (dddd, J_d1_ = 3.7, J_d2_ = 5.5, J_d3_ = 7.3, J_d4_ = 8.1, 1H ^9^CH); 2.13 (ddd, J_d1_ = 6.7, J_d2_ = 8.9, J_d3_ = 13.0, 1H ^4^CH_2_); 2.19 (ddd, J_d1_ = 4.9, J_d2_ = 8.3, J_d3_ = 13.0, 1H ^4^CH_2_); 2.52 (ddd, J_d1_ = 8.5, J_d2_ = 10.2, J_d3_ = 13.1, 1H ^6^CH_2_); 2.53 (dddd, J_d1_ = 2.6, J_d2_ = 4.9, J_d3_ = 8.9, J_d4_ = 18.6, 1H ^3^CH_2_); 2.63 (dddd, J_d1_ = 2.6, J_d2_ = 6.7, J_d3_ = 8.3, J_d4_ = 18.6, 1H ^3^CH_2_); 3.62 (ddd, J_d1_ = 5.5, J_d2_ = 5.8, J_d3_ = 13.7, 1H ^10^CH_2_); 3.65 (ddd, J_d1_ = 3.7, J_d2_ = 7.4, J_d3_ = 13.7, 1H ^10^CH_2_); 4.81 (dd, J_d1_ = 5.8, J_d2_ = 7.4, 1H OH); 6.99 (dd, J_d1_ = 2.6, J_d2_ = 2.6, 1H ^2^CH=). ^13^C NMR (125 MHz, CDCl_3_, δ): 22.87, 25.34, 27.60, 35.55, 36.61 (^3^CH_2_, ^4^CH_2_, ^6^CH_2_, ^7^CH_2_, ^8^CH_2_), 51.22 (^9^CH), 61.84 (^10^CH_2_), 84.39 (^5^C), 137.33 (^2^CH=N).

*[(5R(S),6R(S))-1-Oxido-2-pent-4-en-1-yl-1-azaspiro[4.4]non-1-en-6-yl]methanol* (**12**). To the cooled-on-the-ice-bath solution of triethylamine (5.95 g; 59.0 mmol) and nitrone **9** (7.66 g; 45.3 mmol) in dry THF (40 mL), the solution of Me_3_SiCl (5.88 g, 54.4 mmol) in dry THF (15 mL) was added dropwise. The mixture was stirred at room temperature until the reaction complete (TLC control on silica gel, CHCl_3_:CH_3_OH = 10:1, R_f_ = 0.75, visualization with UV-254). The reaction mixture was concentrated in a vacuum. The crude mixture was diluted with dry Et_2_O (40 mL) and insoluble precipitate was filtered off. The resulting solution contained (*5R,6R)-6-(((trimethylsilyl)oxy)methyl)-1-azaspiro[4.4]non-1-ene 1-oxide* (**10**), ^1^H NMR (400 MHz, CDCl_3_, δ): 0.04 (s, 9H)(^11^CH_3_, ^12^CH_3_, ^13^CH_3_); 1.50–1.64 (m, 1H); 1.69–1.82 (m, 3H); 1.88–2.00 (m, 1H); 2.00–2.11 (m, 1H); 2.14–2.24 (m, 1H); 2.31–2.46 (m, 2H); 2.48–22.70 (m, 2H); 2.67 (d, J_d_ = 7.0, 2H ^10^CH_2_); 6.75 (t, J_t_ = 2.5, 1H ^2^CH=). The solution of crude **10** without further purification was added dropwise to the solution of pent-4-en-1-ylmagnesium bromide which was prepared via slow addition of a 5-bromopentene-1 (10.13 g, 68 mmol) and Et_2_O (15 mL) mixture to a suspension of Mg chips (1.74 g, 72.5 mmol) in dry Et_2_O (15 mL). The reaction mixture was stirred for 2 h, quenched with water (10 mL), and filtered. The filtrate was concentrated in vacuum and the residue was dissolved in MeOH (80 mL). Then, the solution of CuSO_4_·5 H_2_O (10 mg) in 1 mL of NH_3_ (aq.) was added to the mixture, and the air was bubbled until the solution turned dark blue (TLC control on silica gel, CHCl_3_:CH_3_OH = 15:1, R_f_ = 0.4, visualization with UV-254, KMnO_4_ in water). Methanol was evaporated in a vacuum. Then, the mixture was diluted with brine (20 mL) with the addition of 25% aqueous NH_3_ (5 mL) and was extracted with CHCl_3_ (3 × 20 mL). The organic phase was evaporated in a vacuum, and the residue was separated via column chromatography on silica gel (CHCl_3_:CH_3_OH = 15:1) to give **12** as colorless oil. Yield: 9.13 g (85%). UV-Vis (λ_max_ (nm)/lg(ε)): 236 (3.97). IR (neat, cm^−1^): 3310 (O-H), 3076 (H-C=), 1641 (C=C), 1600 (C=N). Analysis: found C 70.90, H 9.77, N 5.78; calcd. for C_14_H_23_NO_2_ C 70.85, H 9.77, N 5.90. HRMS (EI/DFS) *m*/*z* [M^+^]: found: 237.1722; calcd. for C_14_H_23_NO_2_: 237.1723. ^1^H NMR (400 MHz, CDCl_3_, δ): 1.46–1.56 (m, 1H); 1.56–1.66 (m, 3H); 1.70–1.81 (m, 1H); 1.83–2.10 (m, 7H); 2.37–2.67 (m, 5H); 3.52 (ddd, J_d1_ = 2.7, J_d2_ = 4.6, J_d3_ = 12.2, 1H ^10^CH_2_); 3.58 (ddd, J_d1_ = 6.1, J_d2_ = 8.0, J_d3_ = 12.2, 1H ^10^CH_2_); 4.93 (tdd, J_t_ = 1.1, J_d1_ = 1.9, J_d2_ = 10.2, 1H ^15b^CH); 4.98 (tdd, J_t_ = 1.6, J_d1_ = 1.9, J_d2_ = 17.1, 1H ^15a^CH); 5.36 (dd, J_d1_ = 4.6, J_d2_ = 8.0, 1H OH); 5.74 (tdd, J_t_ = 6.7, J_d1_ = 10.2, J_d2_ = 17.1, 1H ^14^CH=). ^13^C NMR (100 MHz, CDCl_3_, δ): 22.72, 24.19, 26.79, 27.16, 28.41, 33.63, 33.92, 36.09 (^3^CH_2_, ^4^CH_2_, ^6^CH_2_, ^7^CH_2_, ^8^CH_2_, ^11^CH_2_, ^12^CH_2_, ^13^CH_2_), 50.99 (^9^CH), 61.87 (^10^CH_2_), 84.45 (^5^C), 115.62 (^15^CH_2_=), 137.52 (^14^CH=), 151.46 (^2^C=N).

*((1R(S),2R(S),6a′R(S),9a′R(S))-Hexahydro-6′H-spiro[cyclopentan-1,3′-cyclopenta[c]pyrrolo[1,2-b]isoxazol]-2-yl)methanol* (**13**). A solution of **12** (0.823 g; 3.42 mmol) and TEMPO (2–3 mg) in toluene (10 mL) was bubbled with Ar and the mixture was heated at 100 °C for 48 h. The resulting solution was concentrated in vacuum, and the residue was separated via column chromatography on silica gel (hexane:EtOAc = 1:1) to give **13** as colorless oil. Yield: 0.75 g (91%). IR (neat, cm^−1^): 3330 (O-H). Analysis: found C 71.15, H 9.78 N 5.65; calcd. for C_14_H_23_NO_2_ C 70.85, H 9.77, N 5.90. HRMS (EI/DFS) *m*/*z* [M^+^]: found: 237.1720; calcd. for C_14_H_23_NO_2_: 237.1723. ^1^H NMR (600 MHz, CDCl_3_, δ): 1.13–1.20 (m, 1H ^8^CH_2_); 1.36–1.43 (m, 2H)(one from ^6^CH_2_ and one from ^7^CH_2_); 1.44–1.50 (m, 1H ^13^CH_2_); 1.51–1.61 (m, 3H)(one from ^4^CH_2_, one from ^11^CH_2_ and one from ^12^CH_2_); 1.61–1.71 (m, 4H one from ^4^CH_2_, one from ^7^CH_2_, one from ^8^CH_2_ and one from ^12^CH_2_); 1.73–1.79 (m, 1H ^13^CH_2_); 1.79–1.84 (m, 1H ^9^CH); 1.84 (ddd, J_d1_ = 6.5, J_d2_ = 9.3, J_d3_ = 12.9, 1H ^3^CH_2_); 1.92 (ddd, J_d1_ = 5.0, J_d2_ = 6.8, J_d3_ = 12.9, 1H ^3^CH_2_); 1.98–2.03 (m, 1H ^11^CH_2_); 2.10–2.18 (m, 1H ^6^CH_2_); 2.45 (ddd, J_d1_ = 3.3, J_d2_ = 7.0, J_d3_ = 12.1, 1H ^14^CH); 3.36 (dd, J_d1_ = 3.9, J_d2_ = 11.4, 1H ^10^CH_2_); 3.50 (dd, J_d1_ = 3.3, J_d2_ = 8.5, 1H ^15^CH_2_); 3.66 (dd, J_d1_ = 10.0, J_d2_ = 11.4, 1H ^10^CH_2_); 3.81 (dd, J_d1_ = 7.0, J_d2_ = 8.5, 1H)(^15^CH_2_); 5.40–5.75 (br. s, 1H OH). ^13^C NMR (150 MHz, CDCl_3_, δ): 21.43 (^7^CH_2_), 25.76 (^12^CH_2_), 25.98 (^8^CH_2_), 30.83 (^6^CH_2_), 32.58 (^13^CH_2_), 34.40 (^3^CH_2_), 38.56 (^4^CH_2_), 39.44 (^11^CH_2_), 48.16 (^9^CH), 55.01 (^14^CH), 65.49 (^10^CH_2_), 76.30 (^15^CH_2_), 80.07 (^5^C), 83.41 (^2^C).

*(1R(S),5R(S),7R(S),8R(S))-6-Azadispiro[4.1.4.2]tridecane-1,8-diyldimethanol* (**14**). Zn powder (1.08 g, 16.62 mmol) was added in one portion to a warm (60 °C) stirred solution containing **13** (0.394 g, 1.66 mmol), EtOH (3 mL), 10 M AcOH (10 mL), and EDTA disodium salt (2.4 g). The reaction mixture was stirred at 60 °C for 1 h and then cooled down to room temperature. The mixture was basified to pH 10 with NaOH solution and extracted with EtOAc (4 × 15 mL). The organic extract was dried with Na_2_CO_3_. After evaporation of the solvent, the crude residue was purified via recrystallisation (hexane:EtOAc = 10:1) to give **14** as colorless crystals. Yield: 0.393 g (99%). M.p. 73–77 °C. IR (KBr, cm^−1^): 3288 (OH, N-H). Analysis: found C 70.39, H 10.28, N 5.87; calcd. for C_14_H_25_NO_2_ C 70.25, H 10.53, N 5.85. HRMS (EI/DFS) *m*/*z* [M^+^]: found: 239.1883; calcd. for C_14_H_25_NO_2_: 239.1880. ^1^H NMR (400 MHz, CDCl_3_, δ): 1.33–1.43 (m, 2H); 1.43–1.55 (m, 4H); 1.55–1.63 (m, 2H); 1.63–1.81 (m, 10H); 3.59 (dd, J_d1_ = 4.4, J_d2_ = 11.0, 2H) (one from ^10^CH_2_ and one from ^15^CH_2_); 3.63 (dd, J_d1_ = 7.0, J_d2_ = 11.0, 2H one from ^10^CH_2_ and one from ^15^CH_2_); 4.16–4.54 (br. s, 2H OH). ^13^C NMR (100 MHz, CDCl_3_, δ): 21.49, 26.53, 37.69, 39.28 (^3^CH_2_ and ^4^CH_2_, ^6^CH_2_ and ^11^CH_2_, ^7^CH_2_ and ^12^CH_2_, ^8^CH_2_ and ^13^CH_2_), 47.88 (^9^CH and ^14^CH), 64.01 (^10^CH_2_ and ^15^CH_2_), 72.32 (^2^C and ^5^C).

*((1R(S),5R(S),7R(S),8R(S))-6-Azadispiro[4.1.4.2]tridecane-1,8-diyl)bis(methylene) diacetate* (**16**). Acetic anhydride (11.73 g; 115 mmol) was added to a solution of **14** (2.75 g; 11.5 mmol) in dry chloroform (40 mL) and the mixture heated under reflux until the reaction was complete (TLC control on silica gel, CHCl_3_:CH_3_OH = 10:1, R_f_ = 0.75, visualization with Dragendorff’s reagent). The reaction mixture was washed with a saturated solution of Na_2_CO_3_ and dried with Na_2_CO_3_. After evaporation of the solvent, the crude residue was separated via column chromatography (silica gel, hexane:EtOAc = 2:1) to give **16** as colorless crystals. Yield: 3.60 g (97%). M.p. 36–37 °C. IR (KBr, cm^−1^): 1739 (C=O). Analysis: found C 66.83, H 9.02, N 4.30; calcd. for C_18_H_29_NO_4_ C 66.84, H 9.04, N 4.33. HRMS (EI/DFS) *m*/*z* [M^+^]: found: 323.2088; calcd. for C_18_H_29_NO_4_: 323.2088. ^1^H NMR (500 MHz, CDCl_3_, δ): 1.34–1.44 (m, 2H); 1.44–1.53 (m, 4H); 1.53–1.67 (m, 6H); 1.69–1.80 (m, 4H); 1.83–1.92 (m, 2H); 1.99 (s, 6H ^17^CH_3_ and ^19^CH_3_); 3.94 (dd, J_d1_ = 7.4, J_d2_ = 11.0, 2H one from ^10^CH_2_ and one from ^15^CH_2_); 4.17 (dd, J_d1_ = 6.1, J_d2_ = 11.0, 2H one from ^10^CH_2_ and one from ^15^CH_2_). ^13^C NMR (125 MHz, CDCl_3_, δ): 21.14 (^17^CH_3_ и ^19^CH_3_), 21.16, 27.85, 37.78, 40.99 (^3^CH_2_ and ^4^CH_2_, ^6^CH_2_ and ^11^CH_2_, ^7^CH_2_ and ^12^CH_2_, ^8^CH_2_ and ^13^CH_2_), 46.48 (^9^CH and ^14^CH), 66.09 (^10^CH_2_ and ^15^CH_2_), 70.88 (^2^C and ^5^C), 171.18 (^16^C=O and ^18^C=O).

*((1R(S),5R(S),7R(S),8R(S))-1,8-Bis(acetyloxymethyl)-6-azadispiro[4.1.4.2]tridecane-6-oxyl* (**17**). A solution of **16** (3.49 g, 10.8 mmol) in dry CHCl_3_ (30 mL) was cooled to ca. -50 °C with liquid nitrogen and MCPBA (2.8 g, 16.2 mmol) was added upon stirring. The mixture was allowed to warm up to room temperature (TLC control on silica gel, hexane:EtOAc = 3:1, R_f_ = 0.6, visualization with UV-254 and Dragendorff’s reagent). The reaction mixture was washed with a saturated solution of Na_2_CO_3_, water, and dried with Na_2_SO_4_. After evaporation of the solvent, the crude residue was purified via column chromatography (silica gel, hexane:EtOAc = 4:1) and recrystallized from hexane to give **17** as yellow crystals. Yield: 3.14 g (86%). m.p. 70.8–70.9 °C. UV-Vis (λ_max_ (nm)/lg(ε)): 241 (3.18). IR (KBr, cm^−1^): 1720 (C=O), 1243 (C-OAc). Analysis: found C 63.59, H 8.94, N 4.14; calcd. for C_18_H_28_NO_5_ C 63.88, H 8.34, N 4.30. HRMS (EI/DFS) *m*/*z* [M^+^]: found: 338.1960; calcd. for C_18_H_28_NO_5_: 338.1958

*(4R(S),5S(R),7R(S),8R(S))-4,8-Bis[(acetyloxy)methyl]-6-azoniadispiro[4.1.4.2]tridec-1-ene-6-oxyl* (**18**). Yield: 0182 g (5%). Yellow oil. UV-Vis (λ_max_ (nm)/lg(ε)): 241 (3.17). IR (neat, cm^−1^): 3054 (H-C=), 1739 (C=O), 1618 (C=C). HRMS (EI/DFS) *m*/*z* [M^+^]: found: 336.1802; calcd. for C_18_H_26_NO_5_: 336.1806. To confirm the structure **18** was reduced with Zn-CF_3_COOH to give *(4R(S),5S(R),7R(S),8R(S))-4,8-Bis[(acetyloxy)methyl]-6-azadispiro[4.1.4.2]tridec-1-ene trifluoroacetate* (**19**). ^1^H NMR (400 MHz, CD_3_OD + CF_3_COOH, δ: 1.60–1.80 (m, 3H); 1.81–2.21 (m, 7H); 2.01 and 2.04 (s, both 3H ^17^CH_3_ and ^19^CH_3_); 2.28–2.43 (m, 2H ^14^CH and one from ^8^CH_2_); 2.52–2.62 (m, 1H ^8^CH_2_); 2.69–2.78 (m, 1H ^9^CH); 4.13–4.16 (m, 2H ^15^CH_2_); 4.32 (dd, J_d1_ = 8.1, J_d2_ = 12.0, 1H ^10^CH_2_); 4.39 (dd, J_d1_ = 6.3, J_d2_ = 12.0, 1H ^10^CH_2_); 5.93–5.96 (m, 1H ^6^CH=); 6.14–6.18 (m, 1H ^7^CH=). ^13^C NMR (100 MHz, CD_3_OD + CF_3_COOH, δ): 20.70 and 20.77 (^17^CH_3_ and ^19^CH_3_), 27.31 (^13^CH_2_), 35.53 (^8^CH_2_), 20.90, 36.23, 37.03, 37.17 (^3^CH_2_, ^4^CH_2_, ^11^CH_2_, ^12^CH_2_), 46.41 (^9^CH), 47.37 (^14^CH), 63.82 (^10^CH_2_), 64.74 (^15^CH_2_), 77.16 (^2^C), 81.66 (^5^C), 132.27 (^6^CH=), 137.88 (^7^CH=), 172.23 and 172.46 (^16^C=O and ^18^C=O).

*Dimethyl (1R(S),5R(S),7R(S),8R(S))-6-azadispiro[4.1.4.2]tridecane-1,8-dicarboxylate* (**21**). CrO_3_ (2.16 g, 21.6 mmol) was dissolved in H_2_SO_4_ (4.31 g, 41.1 mmol) and H_2_O (33 mL). The resulting mixture was added dropwise to a stirring cooled in an ice bath solution of **14** (1.0 g, 4.18 mmol) in acetone (20 mL). The mixture was stirred at room temperature until the reaction was complete (20 h. TLC control on silica gel, CH_3_OH:EtOAc = 1:2, visualization with I_2(vap_._)_). Acetone was evaporated and Ba(OH)_2_ water solution was added to the residue to pH = 6. The precipitate formed was filtered off, washed with water and EtOH. The filtrate was evaporated to dryness under reduced pressure and extra pumped out on a high vacuum pump to give crude *(1R(S),5R(S),7R(S),8R(S))-6-azadispiro[4.1.4.2]tridecane-1,8-dicarboxylic acid* (**20**). A small portion of **20** was isolated using column chromatography on silica gel, eluted with gradient from EtOAc:MeOH = 4:1 to pure MeOH. The pure fractions were collected, evaporated, triturated in the mixture CHCl_3_:CCl_4_ = 1:2 and crystallized from the mixture MeOH:EtOAc = 50:1. M.p. 190.6–191.9 °C. IR (KBr, cm^−1^): 3423, 2765, 2700, 2588, 2514, 2468 (O-H, N-H), 1654 (C=O). Analysis: found C 62.91, H 7.89, N 5.31; calcd. for C_14_H_21_NO_4_ C 62.90, H 7.92, N 5.24. ^1^H NMR (400 MHz, CD_3_OD, δ): 1.77–1.97 (m, 6H); 1.97–2.10 (m, 4H); 2.15–2.35 (m, 6H); 2.88 (t, J_t_ = 9.0, 2H ^9^CH and ^14^CH). ^13^C NMR (75 MHz, CD_3_OD, δ): 20.77, 28.50, 34.88, 36.45 (^4^CH_2_ and ^3^CH_2_, ^6^CH_2_ and ^11^CH_2_, ^7^CH_2_ and ^12^CH_2_, ^8^CH_2_ and ^13^CH_2_), 50.20 (^9^CH and ^14^CH), 74.64 (^2^C and ^5^C), 177.89 (^10^COOH and ^15^COOH). The crude **20** was dissolved in methanol (50 mL), saturated with gaseous HCl and left for 5 days at room temperature to complete the reaction (TLC control on silica gel, CH_3_OH:EtOAc = 1:5, visualization with Dragendorff’s reagent). The methanol was distilled off in vacuum and brine (20 mL) was added to the residue, the mixture was basified to pH 10–11 with Na_2_CO_3_ solution and extracted with EtOAc (3 × 20 mL). The organic extract was dried with Na_2_SO_4_. After evaporation of the solvent, the crude residue was purified via column chromatography (silica gel, hexane:EtOAc = 4:1) to give **21** as colorless oil. Yield: 0.778 g (63%). IR (neat, cm^−1^): 1729 (C=O). Analysis: found C 64.90, H 8.76, N 4.81; calcd. for C_16_H_25_NO_4_ C 65.06, H 8.53, N 4.74. HRMS (EI/DFS) *m*/*z* [M^+^]: found: 295.1777; calcd. for C_16_H_25_NO_4_: 295.1778. ^1^H NMR (300 MHz, CDCl_3_, δ): 1.37–1.51 (m, 4H); 1.60–1.73 (m, 6H); 1.73–1.82 (m, 2H); 1.83–2.01 (m, 4H); 2.57 (dd, J_d1_ = 7.5 J_d2_ = 8.5, 2H ^9^CH and ^14^CH); 3.61 (s, 6H ^16^CH_3_ and ^17^CH_3_). ^13^C NMR (75 MHz, CDCl_3_, δ): 22.24, 27.52, 37.59, 41.02 (^3^CH_2_ and ^4^CH_2_, ^6^CH_2_ and ^11^CH_2_, ^7^CH_2_ and ^12^CH_2_, ^8^CH_2_ and ^13^CH_2_), 51.09 (^9^CH and ^14^CH), 53.63 (^16^CH_3_ and ^17^CH_3_), 72.50 (^2^C and ^5^C), 175.03 (^10^C=O and ^15^C=O).

*(1R(S),5R(S),7R(S),8R(S))-1,8-Bis(methoxycarbonyl)-6-azadispiro[4.1.4.2]tridecane-6-oxyl* (**22**). A solution of **21** (2.98 g, 10.8 mmol) in dry CHCl_3_ (30 mL) was cooled in liquid nitrogen to ca. −50 °C and MCPBA (2.17 g, 12.6 mmol) was added upon stirring. The mixture was allowed to warm up to room temperature (TLC control on silica gel, hexane:EtOAc = 3:1, R_f_ = 0.4, visualization with UV-254 and Dragendorff’s reagent). The reaction mixture was washed with a saturated solution of Na_2_CO_3_, water and dried with Na_2_SO_4_. After evaporation of the solvent, the crude residue was purified via column chromatography (silica gel, hexane:EtOAc = 5:1) and recrystallized from hexane:Et_2_O = (10:1) mixture to give **22** as yellow crystals. Yield: 2.6 g (83%). M.p. 84.4–88.8 °C. UV-Vis (λ_max_ (nm)/lg(ε)): 241 (3.26). IR (KBr, cm^−1^): 1731 (C=O). Analysis: found C 61.88, H 7.72, N 4.58; calcd. for C_16_H_24_NO_5_ C 61.92, H 7.79, N 4.51. HRMS (EI/DFS) *m*/*z* [M^+^]: found: 310.1649; calcd. for C_16_H_24_NO_5_: 310.1646. ^1^H NMR (300 MHz, CD_3_OD + CF_3_COOH, δ): 1.80–2.00 (m, 6H); 2.00–2.37 (m, 10H); 3.15 (dd, J_d1_ = 7.1 J_d2_ = 8.6, 2H ^9^CH and ^14^CH); 3.81 (s, 6H ^16^CH_3_ and ^17^CH_3_).

*1,8-Dicarboxy-6-azadispiro[4.1.4.2]tridecane-6-oxyl* (**23a–c**) (mixture of isomers). A solution of KOH (10%) in water:methanol = 1:1mixture (8 mL) was added to a solution of **22** (0.11 g, 0.32 mmol) in MeOH (5 mL) and allowed to stand at r.t. until the reaction was complete (up to 24 h). The progress of the reaction was monitored by TLC (silica gel, EtOAc:AcOH = 100:1, visualization with UV-254). The solution was evaporated under reduced pressure. The residue was dissolved in water, the pH was adjusted to 3 with NaHSO_4_ 10% solution and the mixture was extracted with EtOAc (3 × 8 mL). The extract was dried with Na_2_SO_4_ and evaporated to give **23a-c** as a yellow powder with 1:2:1 ratio of isomers according to ^1^H NMR after reduction with Zn/CF_3_COOH (see Appendix A).

*(1R(S),5R(S),7R(S),8R(S))-1,8-Dicarboxy-6-azadispiro[4.1.4.2]tridecane-6-oxyl* (**23a**). A solution of ascorbic acid (0.109 g, 6.21 mmol) in water (3 mL) was stirred under argon and pH was adjusted to ca. 5 with Na_2_CO_3_. Then, a solution of **22** (0.175 g, 5.65 mmol) in methanol (2 mL) was added and the mixture was stirred for 1.5 h. The resulting mixture was extracted with Et_2_O (3 × 1 mL) under argon via injecting, mixing, and taking the upper layer with a syringe. The extract was placed into separate vial and ether was removed in the stream of argon. A solution of KOH (10%) in water–methanol mixture 1:1, (8 mL) was added and the solution was allowed to stand at room temperature for 48 h under argon. Then, air was bubbled through the reaction mixture for 8 h. The pH was adjusted to 3 with NaHSO_4_ 10% solution and the mixture was extracted with EtOAc (3 × 2 mL). The extract was dried with Na_2_SO_4_ and concentrated in vacuum; the residue was separated via column chromatography (silica gel, hexane:EtOAc:AcOH = 10:100:1) to give pure **23a** as yellow crystals. Total yield: 0.072 g (45%). M.p. 152–153 °C. UV-Vis (λ_max_ (nm)/lg(ε)): 239 (3.23). IR (KBr, cm^−1^): 3400, 3078, 2669, 2570 (O-H), 1704 (C=O). Analysis: found C 59.58, H 6.90, N 4.87; calcd. for C_14_H_20_NO_5_ C 59.56, H 7.14, N 4.96. HRMS (EI/DFS) *m*/*z* [M^+^]: found: 282.1333; calcd. for C_14_H_20_NO_5_: 282.1336. ^1^H NMR (500 MHz, CD_3_OD + CF_3_COOH, δ): 1.81–1.96 (m, 6H); 2.04–2.14 (m, 4H); 2.16–2.23 (m, 2H); 2.23–2.32 (m, 4H); 3.06 (t, J_t_ = 8.3, 2H ^9^CH and ^14^CH). ^13^C NMR (125 MHz, CD_3_OD + CF_3_COOD, δ): 21.79, 29.38, 36.08, 37.28 (^3^CH_2_ and ^4^CH_2_, ^6^CH_2_ and ^11^CH_2_, ^7^CH_2_ and ^12^CH_2_, ^8^CH_2_ and ^13^CH_2_), 50.46 (^9^CH and ^14^CH), 76.22 (^2^C and ^5^C), 178.53 (^10^C=O and ^15^C=O).

*(1R(S),5R(S),7R(S),8R(S))-1,8-Bis(((1H-imidazol-1-ylcarbonyl)oxy)methyl)-6-azadispiro[4.1.4.2]tridecane-6-oxyl* (**26**). The solution of CDI (0.359 g, 2.216 mmol) in dry THF (3 mL) was added dropwise to the solution of **5** (0.268 g; 1.055 mmol) in dry THF (4 mL). The mixture was stirred at room temperature until the reaction was complete (about 12 h. TLC control on silica gel, hexane:Et_2_O = 1:2, visualization with UV-254). The reaction mixture was diluted with dry Et_2_O and cooled in a refrigerator for 2 h. The precipitate formed was filtered off, washed with dry Et_2_O, dried in air, and recrystallized from CH_2_Cl_2_. Yellow crystals. Yield: 0.448 g (96%). M.p. 128.5–128.8 °C. UV-Vis (λ_max_ (nm)/lg(ε)): 230 (3.95). IR (KBr, cm^−1^): 3149, 3130, 3112, 3101 (H-C=), 1766 (C=O). Analysis: found C 60.01, H 5.99, N 15.87; calcd. for C_22_H_28_N_5_O_5_ C 59.72, H 6.38, N 15.83. HRMS (EI/DFS) *m*/*z* [M^+^]: found: 442.2087; calcd. for C_22_H_28_N_5_O_5_: 442.2085.

*(1R(S),5R(S),7R(S),8R(S))-1,8-Bis[(([{3-(dimethylamino)propyl}amino]carbonyl)oxy)methyl]-6-azadispiro[4.1.4.2]tridecane-6-oxyl* (**27**). *N*,*N*-Dimethylpropane-1,3-diamine (0.217 g, 2.12 mmol) was added to the solution of **26** (0.313 g; 0.708 mmol) in dry CH_2_Cl_2_ (6 mL). The mixture was stirred at room temperature until the reaction was complete (about 100 h. TLC control on silica gel, hexane:Et_2_O = 1:2, visualization with UV-254). The reaction mixture was evaporated in a vacuum, and the residue was separated via column chromatography on Al_2_O_3_ (CH_2_Cl_2_:CH_3_OH = 10:1) to give **27** as yellow oil. Yield: 0.199 g (55%). IR (neat, cm^−1^): 3332 (N-H), 1714, 1699 (C=O). Analysis: found C 61.00, H 9.47, N 13.91; calcd. for C_26_H_48_N_5_O_5_ C 61.15, H 9.70, N 13.71. HRMS (EI/DFS) *m*/*z* [M^+^]: found: 510.3648; calcd. for C_26_H_48_N_5_O_5_: 510.3650. ^1^H NMR (500 MHz, CD_3_OD + CF_3_COOD, δ): 1.70–1.84 (m, 4H); 1.85–2.03 (m, 10H); 2.03–2.15 (m, 4H); 2.16–2.26 (m, 2H); 2.36–2.43 (m, 2H ^9^CH and ^14^CH); 2.89 (s, 12H ^20^CH_3_, ^21^CH_3_, ^26^CH_3_, ^27^CH_3_); 3.13–3.19 (m, 4H ^17^CH_2_ and ^23^CH_2_); 3.23 (t, J_t_ = 6.5, 4H ^19^CH_2_ and ^25^CH_2_); 4.22 (dd, J_d1_ = 5.4 J_d2_ = 11.9, 2H one from ^10^CH_2_ and one from ^15^CH_2_); 4.28 (dd J_d1_ = 6.3 J_d2_ = 11.9, 2H one from ^10^CH_2_ and one from ^15^CH_2_). ^13^C NMR (125 MHz, CD_3_OD + CF_3_COOD, δ): 20.92, 26.19, 27.33, 36.41, 37.22, 38.61 (^3^CH_2_ and ^4^CH_2_, ^6^CH_2_ and ^11^CH_2_, ^7^CH_2_ and ^12^CH_2_, ^8^CH_2_ and ^13^CH_2_, ^17^CH_2_ and ^23^CH_2_, ^18^CH_2_ and ^24^CH_2_), 43.42 and 43.47 (^20^CH_3_, ^21^CH_3_, ^26^CH_3_, ^27^CH_3_), 47.68 (^9^CH and ^14^CH), 56.55 (^19^CH_2_ and ^25^CH_2_), 65.26 (^10^CH_2_ and ^15^CH_2_), 77.62 (^2^C and ^5^C), 158.75 (^16^C=O and ^22^C=O).

*1R(S),5R(S),7R(S),8R(S)-1,8-Bis[(([{3-methoxy-3-oxopropyl}amino]carbonyl)oxy)methyl]-6-azadispiro[4.1.4.2]tridecane-6-oxyl* (**28**). Methyl 3-isocyanatopropanoate (0.754 g, 5.846 mmol) was added to the solution of **5** (0.675 g; 2.567 mmol) in dry THF (8 mL). The mixture was heated to reflux until the reaction was complete (ca. 72 h, TLC control on silica gel, hexane:Et_2_O = 1:5, visualization with UV-254). The reaction mixture was evaporated in a vacuum, and the residue was purified via column chromatography (silica gel, Et_2_O) to give **28** as yellow crystals. Yield: 1.35 g (99%). M.p.: 74.2–80.1 °C. IR (neat, cm^−1^): 3349 (N-H), 1735, 1720, (C=O). Analysis: found C 56.10, H 7.50, N 8.09; calcd. for C_24_H_38_N_3_O_9_ C 56.24, H 7.47, N 8.20. ^1^H NMR (500 MHz, CD_3_OD + CF_3_COOD, δ): 1.70–1.85 (m, 4H); 1.87–2.23 (m, 12H); 2.36–2.44 (m, 2H ^9^CH and ^14^CH); 2.56 (t, J_t_ = 6.5, 4H ^18^CH_2_ and ^23^CH_2_); 3.42 (t, J_t_ = 6.5, 4H ^17^CH_2_ and ^22^CH_2_); 3.68 (s, 6H ^20^CH_3_ and ^25^CH_3_); 4.21–4.30 (m, 4H ^10^CH_2_ and ^15^CH_2_). ^13^C NMR (125 MHz, CD_3_OD + CF_3_COOD, δ): 20.91, 27.41, 34.98, 36.58, 36.86, 37.76 (^3^CH_2_ and ^4^CH_2_, ^6^CH_2_ and ^11^CH_2_, ^7^CH_2_ and ^12^CH_2_, ^8^CH_2_ and ^13^CH_2_, ^17^CH_2_ and ^22^CH_2_, ^18^CH_2_ and ^23^CH_2_), 47.86 (^9^CH and ^14^CH), 52.21 (^20^CH_3_ and ^25^CH_3_), 65.35 (^10^CH_2_ and ^15^CH_2_), 77.37 (^2^C and ^5^C), 158.50 (^16^C=O and ^21^C=O), 173.90 (^19^C=O and ^24^C=O).

*1R(S),5R(S),7R(S),8R(S)-1,8-Bis[(([{2-carboxyethyl}amino]carbonyl)oxy)methyl]-6-azadispiro[4.1.4.2]tridecane-6-oxyl* (**29**). An aqueous solution of NaOH (1%, 100 mL) was added to a solution of nitroxide **28** (1.21 g, 2.36 mmol) in MeOH (100 mL), and the mixture was allowed to stand at room temperature until the reaction was complete (ca. 24 h). The progress of the reaction was monitored by TLC (silica gel, EtOAc, visualization with UV-254). The methanol was distilled off under reduced pressure and the pH was adjusted to 4 with the saturated solution of NaHSO_4_ and extracted with EtOAc (5 × 15 mL). The organic phase was concentrated in a vacuum, and the residue was separated via column chromatography on silica gel (EtOAc:AcOH = 50:1) to give **29** as yellow crystals. Yield: 1.05 g (92%). M.p.: 116–119 °C. IR (KBr, cm^−1^): 3369 (N-H, O-H), 1718 (C=O). Analysis: found C 54.10, H 6.87, N 8.63; calcd. for C_22_H_34_N_3_O_9_ C 54.54, H 1.07, N 8.67. ^1^H NMR (400 MHz, CD_3_OD + CF_3_COOD, δ): 1.60–1.78 (m, 4H); 1.78–2.15 (m, 12H); 2.28–2.37 (m, 2H ^9^CH and ^14^CH); 2.47 (t, J_t_ = 6.5, 4H ^18^CH_2_ and ^22^CH_2_); 3.34 (t, J_t_ = 6.5, 4H ^17^CH_2_ and ^21^CH_2_); 4.19 (d, J_d_ = 5.5, 4H ^10^CH_2_ and ^15^CH_2_). ^13^C NMR (150 MHz, CD_3_OD + CF_3_COOD, δ): 20.96, 27.47, 34.99, 36.63, 36.99, 37.83 (^3^CH_2_ and ^4^CH_2_, ^6^CH_2_ and ^11^CH_2_, ^7^CH_2_ and ^12^CH_2_, ^8^CH_2_ and ^13^CH_2_, ^17^CH_2_ and ^21^CH_2_, ^18^CH_2_ and ^22^CH_2_), 47.88 (^9^CH and ^14^CH), 65.36 (^10^CH_2_ and ^15^CH_2_), 77.47 (^2^C and ^5^C), 158.45 (^16^C=O and ^20^C=O), 175.37 (^19^COOH and ^23^COOH).

*(1R(S),5R(S),7R(S),8R(S))-1,8-Bis[(prop-2-yn-1-yloxy)methyl]-6-azadispiro[4.1.4.2]tridecane-6-oxyl* (**30**) and *(1R(S),5R(S),7R(S),8R(S))-1-(Hydroxymethyl)-8-[(prop-2-yn-1-yloxy)methyl]-6-azadispiro[4.1.4.2]tridecane-6-oxyl* (**31**). To the solution of **5** (1.015 g; 3.99 mmol) in dry THF (16 mL), the NaH (50% suspension in oil, 0.228 g, 5.99 mmol) was added. The mixture was stirred at room temperature for 1 h. Then, the solution of propargyl bromide in toluene (80%, 0.713 g; 5.99 mmol) was added. The mixture was stirred at room temperature until the reaction was complete (about 12 h. TLC control on silica gel, hexane:Et_2_O = 3:4, visualization with UV-254). The mixture was diluted with AcOH (pH = 5–6) and Et_2_O (15 mL) and washed with brine (2 × 15 mL). The organic phase was separated, and after evaporation of the solvent, the crude residue was separated using column chromatography on silica gel (hexane:Et_2_O from 3:1 to 1:4) to give **30** and **31. 30**: yellow crystals, yield **30** 0.475 g (36%), m.p.: 54.1–54.9 °C. UV-Vis (λ_max_ (nm)/lg(ε)): 240 (3.26). IR (KBr, cm^−1^): 3293, 3251 (H-C≡), 2113 (C≡C). Analysis: found C 73.11, H 8.96, N 4.27; calcd. for C_20_H_28_NO_3_ C 73.69, H 8.54, N 4.24. HRMS (EI/DFS) *m*/*z* [M^+^]: found: 330.2062; calcd. for C_20_H_28_NO_3_: 330.2064. **31**: yellow oil, yield **31** 0.467 g (40%). UV-Vis (λ_max_ (nm)/lg(ε)): 232 (3.17). IR (neat, cm^−1^): 3421 (O-H), 3305, 3253 (H-C≡), 2111 (C≡C). Analysis: found C 69.60, H 8.64, N 4.58; calcd. C 69.83, H 8.96, N 4.79. HRMS (EI/DFS) *m*/*z* [M^+^]: found: 292.1907; calcd. for C_17_H_26_NO_3_: 292.1902.

*(1R(S),5R(S),7R(S),8R(S))-1-(Hydroxymethyl-8-({(1-[2,3,4,6-tetra-O-acetyl-β-d-galactopyranosyl]-1H-1,2,3-triazol-4-yl)methoxy}methyl)-6-azadispiro[4.1.4.2]tridecane-6-oxyl* (**32**). The solution of 2,3,4,6-tetra-*O*-acetyl-β-*D*-galactopyranosyl azide (0.344 g, 0.921 mmol) in EtOH (4 mL) was added dropwise to the solution of **31** (0.269 g; 0.921 mmol) in EtOH (2 mL). A solution of CuSO_4_ in H_2_O (10%, 300 μL, 0.19 mmol) was mixed with a solution of ascorbic acid (190 mg, 1.08 mmol) in water (2 mL) and immediately poured into the above solution of azide and alkyne. The mixture was stirred at room temperature until the reaction was complete (ca. 12 h, TLC control on silica gel, hexane:Et_2_O = 1:2, visualization with UV-254). EtOH was distilled off in vacuum, brine (8 mL) was added to the residue and the mixture was extracted with CHCl_3_ (2 × 15 mL). The organic phase was evaporated in a vacuum, and the residue was separated via column chromatography on silica gel (CHCl_3_:CH_3_OH = 20:1) to give **32** as yellow glassy solid. Yield: 10.21 g (68%). IR (KBr, cm^−1^): 3434 (O-H), 3145 (H-C=), 1755 (C=O), 1226 (C-OAc). Analysis: found C 56.07, H 6.80, N 8.07; calcd. for C_31_H_45_N_4_O_12_ C 55.93, H 6.81, N 8.42. ^1^H NMR (400 MHz, CD_3_OD + CF_3_COOD, δ): 1.55–2.25 (m, 17H); 1.87, 1.97, 2.01, 2.20 (s, all 3H ^29^CH_3_, ^30^CH_3_, ^31^CH_3_, ^32^CH_3_); 2.27–2.37 (m, 1H); 3.71 (ddd, J_d1_ = 4.2 J_d2_ = 7.3 J_d3_ = 10.3, 1H ^10^CH_2_ or ^15^CH_2_); 3.81 (ddd, J_d1_ = 6.6 J_d2_ = 6.8 J_d3_ = 11.5, 1H ^10^CH_2_ or ^15^CH_2_); 3.86 (ddd, J_d1_ = 3.6 J_d2_ = 10.2 J_d3_ = 10.3, 1H ^10^CH_2_ or ^15^CH_2_); 3.94 (ddd, J_d1_ = 0.9 J_d2_ = 3.5 J_d3_ = 11.5, 1H ^10^CH_2_ or ^15^CH_2_); 4.15 (ddd, J_d1_ = 0.7 J_d2_ = 7.0 J_d3_ = 11.4, 1H); 4.24 (dd, J_d1_ = 5.6 J_d2_ = 11.4, 1H); 4.50 (tdd, J_t_ = 0.9 J_d1_ = 6.0 J_d2_ = 6.9, 1H); 4.71 (m, 2H); 5.44 (ddd, J_d1_ = 1.0 J_d2_ = 3.4 J_d3_ = 10.3, 1H); 5.56–5.63 (m, 2H); 6.11 (dd, J_d1_ = 1.9 J_d2_ = 9.1, 1H ^19^CH); 8.29 (d, J_d_ = 5.4, 1H ^18^CH=). ^13^C NMR (100 MHz, CD_3_OD + CF_3_COOD, δ): 20.15, 20.37, 20.41, 20.46 (^29^CH_3_, ^30^CH_3_, ^31^CH_3_, ^32^CH_3_), 21.16, 21.51, 26.22, 26.55, 35.77, 35.81, 36.00, 36.86, 36.96, 37.70 (^3^CH_2_, ^4^CH_2_, ^6^CH_2_, ^7^CH_2_, ^8^CH_2_, ^11^CH_2_, ^12^CH_2_, ^13^CH_2_), 46.32 and 47.31 (^9^CH and ^14^CH), 62.12, 62.59, [64.74 and 64.82], [70.64 and 70.72], [77.76 and 77.87], [78.81 and 78.82], 68.58, 69.75, 72.10, 74.99 (^20^CH, ^21^CH, ^22^CH, ^23^CH), 87.09 (^19^CH), 124.41 (^18^CH=), 145.23 (^17^C), 170.67, 171.36, 171.89, 172.15 (^25^C=O, ^26^C=O, ^27^C=O, ^28^C=O).

*(1R(S),5R(S),7R(S),8R(S))-1-(Hydroxymethyl)-8-({(1-[β-d-galactopyranosyl]-1H-1,2,3-triazol-4-yl)methoxy}methyl)-6-azadispiro[4.1.4.2]tridecane-6-oxyl* (**33**). Nitroxide **32** (0.26 g; 0.39 mmol) was dissolved in saturated solution of NH_3_ in MeOH (5 mL) and allowed to stand at r.t. until the reaction was complete (up to 72 h). The progress of the reaction was monitored by TLC (silica gel, CHCl_3_:MeOH = 7:1, visualization with UV-254). The solution was evaporated under reduced pressure and the residue was separated using column chromatography on silica gel (SiO_2_, CHCl_3_:CH_3_OH from 3:1 to 2:1) to give **33** as yellow glassy solid. Yield: 0.154 g (79%). IR (neat, cm^−1^): 3440 (O-H). Analysis: found C 55.87, H 7.80, N 11.07; calcd. for C_23_H_37_N_4_O_8_ C 55.52, H 7.50, N 11.26. ^1^H NMR (500 MHz, CD_3_OD + CF_3_COOD, δ): 1.53–1.72 (m, 4H); 1.74–2.08 (m, 9H); 2.09–2.23 (m, 4H); 2.31–2.38 (m, 1H); 3.75–3.84 (m, 5H); 3.88–3.96 (m, 3H); 4.04–4.08 (m, 1H); 4.17–4.23 (m, 1H); 4.73 (d, J_d_ = 12.5, 1H ^16^CH_2_); 4.73 (d, J_d_ = 12.5, 1H ^16^CH_2_); 5.65 (d, J_d_ = 9.2, 1H ^19^CH); 8.29–8.31 (m, 1H ^18^CH=). ^13^C NMR (125 MHz, CD_3_OD + CF_3_COOD, δ): [21.20 and 21.22], 21.42, 26.15, 26.54, 35.73, [35.94 and 35.97], 36.88, [37.57 and 37.59] (^3^CH_2_, ^4^CH_2_, ^6^CH_2_, ^7^CH_2_, ^8^CH_2_, ^11^CH_2_, ^12^CH_2_, ^13^CH_2_), 46.23 and 47.19 (^9^CH and ^14^CH), 62.14, 62.30, 64.82, 70.89 (^10^CH_2_, ^15^CH_2_, ^16^CH_2_, ^24^CH_2_), 77.78 and 78.75 (^2^C и ^5^C), 70.22, 71.33, 75.10, 79.70 (^20^CH, ^21^CH, ^22^CH, ^23^CH), 89.95 (^19^CH), [124.26 and 124.31] (^18^CH), [144.75 and 144.78] (^17^C).

*2,2,5,5-Tetramethyl-3-(((1-((β-d-galactopyranosyl)-1H-1,2,3-triazol-4-yl)methoxy)methyl)pyrrolidine-1-oxyl* (**35**) was prepared according to the Figure 9.

*2,2,5,5-Tetramethyl-3-((prop-2-ynyloxy)methyl)pyrrolidine-1-oxyl (***37***)*. Sodium hydride (50% in oil, 170 mg, 3.54 mmol) was added portion-wise under argon to a stirred solution of **36** (580 mg, 3.37 mmol) in dry THF (5 mL). The mixture was stirred at room temperature under argon for 1 h; then, the solution of propargyl bromide in toluene (80%, 0.5 g, 4.2 mmol) was added. The mixture was stirred at room temperature overnight, quenched with HOAc (0.5 mL), diluted with brine (5 mL), the resulting nitroxide was extracted with ether:hexane = 1:1 mixture and the extract was dried with MgSO_4_. The solution was concentrated in vacuum and the residue was separated using column chromatography on silica gel (CH_2_Cl_2_) to give **37** as a yellow oil. Yield 0.45 g (64%). IR (KBr, cm^−1^): 3289, 3232 (H-C≡), 2112 (C≡C). Analysis: found C 68.46, H 9.55, N 6.67; calcd. for C_12_H_20_NO_2_ C 68.54, H 9.59, N 6.66. HRMS (EI/DFS) *m*/*z* [M^+^]: found: 210.1488; calcd. for C_12_H_20_NO_2_: 210.1489.

*2,2,5,5-Tetramethyl-3-(((1-(2,3,4,6-tetra-O-acetyl-β-d-galactopyranosyl)-1H-1,2,3-triazol-4-yl)methoxy)methyl)pyrrolidine-1-oxyl (***38***)*. 2,3,4,6-Tetra-*O*-acetyl-β-d-galactopyranosyl azide (0.570 g, 1.53 mmol) and 2,2,5,5-tetramethyl-3-((prop-2-ynyloxy)methyl)pyrrolidine 1-oxyl (330 mg, 1.57 mmol) were dissolved in a mixture of EtOH (5 mL) and H_2_O (1 mL) at 40 °C. A solution of CuSO_4_ in H_2_O (10%, 300 μL, 0.19 mmol) was mixed with a solution of ascorbic acid (100 mg, 0.57 mmol) in water (1 mL) and immediately poured into the above solution of azide and alkyne. The mixture was left overnight; then, MnO_2_ (0.5 g, 5.7 mmol) was added, the mixture was stirred for 0.5 h, the precipitate was filtered off, washed with EtOH and CHCl_3_, the combined filtrates were concentrated in vacuum and the residue was separated using column chromatography on silica gel (CHCl_3_ + 2% MeOH), and fractions were evaporated in vacuum and triturated with dry diethyl ether to give **38** as a yellow glassy solid, which was dried in high vacuum. Yield 0.79 g (89%). IR (KBr, cm^−1^): 3143 (H-C=), 1755 (C=O), 1227 (C-OAc). Analysis: found C 53.11, H 6.96, N 6.27; calcd. for C_26_H_39_N_4_O_11_ C 53.51, H 6.74, N 9.60. ^1^H NMR (300 MHz, CD_3_OD + CF_3_COOH, δ): 1.34, 1.48, 1.51, 1.53 (each s, 3H)(^1^CH_3_, ^2^CH_3_, ^3^CH_3_, ^4^CH_3_); 1.89, 2.01, 2.04, 2.24 (each s, 3H ^5^CH_3_, ^6^CH_3_, ^7^CH_3_, ^8^CH_3_); 1.88 (m, 1H) and 2.09 (m, 1H) (^9^CH_2_); 2.58 (ddd, J_d1_ = 6.7 J_d2_ = 6.6 J_d3_ = 12.4, 1H ^10^CH); 3.54 (m, 2H ^11^CH_2_); 4.64 (m, 2H ^12^CH_2_); 4.21 (m, 2H ^19^CH_2_); 4.35 (t, J = 6.4, 1H ^18^CH); 5.38 (dd, J_d1_ = 3.1 J_d2_ = 10.2, 1H ^16^CH); 5.57 (m, 2H ^14^CH and ^15^CH); 6.0 (m, 1H ^17^CH); 8.07 (m, 1H ^13^CH).

*2,2,5,5-Tetramethyl-3-(((1-(β-d-galactopyranosyl)-1H-1,2,3-triazol-4-yl)methoxy)methyl)pyrrolidine-1-oxyl (***35***).* The nitroxide **38** (0.6 g, 1.03 mmol) was dissolved in the saturated solution of NH_3_ in dry methanol (25 mL). The solution was left overnight; then, evaporated in vacuum and the residue was separated using column chromatography on silica gel (CHCl_3_ + 20% MeOH). The pure fractions were collected, evaporated in vacuum and the residue was triturated with diethyl ether and dried in high vacuum to give **35** as light-yellow glassy powder. Yield 0.32 g (77%). IR (KBr, cm^−1^): 3410 (O-H), 3153 (H-C=). Analysis: found C 52.11, H 7.96, N 13.27; calcd. for C_18_H_31_N_4_O_7_ C 52.04, H 7.52, N 13.49. ^1^H NMR (300 MHz, CD_3_OD, reduced with Zn/NH_4_Cl at 5 °C, and then, filtered and acidified with few drops of CF_3_COOH, a mixture of invertamers and diastereomers, δ): 1.2–1.6 (m, 12H CH_3_ groups); 1.88 (m), 1.99 (m), 2.11 (m), 2.30 (m), 2.46 (m) and 2.65 (m), (total 3H pyrrolidine ring, two invertamers); 3.63 (m, 2H OCH_2_-); 4.66 (m, 2H CH_2_-C=); 3.7–3.84 (m, 3H), 3.89 (m, 1H), 4.03 (m, 1H), 4.18 (m, 1H), 5.62 (d, J = 8.8 Hz) (Galactose); 8.28 (s, 1H CH=).

## 4. Conclusions

In this work, we described a new family of dispirocyclic nitroxides that show very attractive properties for SDSL/EPR studies in biological media and in cells. The unique feature of these radicals is the combination of high resistance to reduction with improved spin relaxation characteristics typical of nitroxides with two spirocyclic moieties adjacent to N-O^•^ group. These nitroxides can be prepared from commercially available chemicals using a set of simple procedures with an overall yield over 40%. The resulting dispirocyclic core contains two hydroxymethyl groups near the radical center, which opens up the possibility of synthesizing mono- and bifunctional spin labels. It is important that binding to large biomolecules may further increase the resistance to reduction due to proximity of the functional groups to nitroxide moiety.

## Data Availability

All the data are in the text and the Appendix A in this article.

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
