# Peer review of "Synthesis and Properties of (1R(S),5R(S),7R(S),8R(S))-1,8-Bis(hydroxymethyl)-6-azadispiro[4.1.4.2]tridecane-6-oxyl: Reduction-Resistant Spin Labels with High Spin Relaxation Times"

_ijms, 2023, doi:10.3390/ijms241411498_

Round 1

Reviewer 1 Report

As reported by I.A. Kiriliuk and co-workers, site-directed spin labeling coupled to electron paramagnetic resonance spectroscopy (SDSL-EPR) is a powerful biophysical technique for studying biomolecules in physiological environments. It enables the identification of biologically important conformations of membrane proteins in particular. Nitroxides are the most widely used spin markers, and have the advantage of being able to be used both to study the local dynamics of biomolecules at physiological temperatures and to measure inter-spin distances. In this work, I.A.Kiriliuk and co-workers are investigating new dispirocyclic nitroxides with interesting properties. These radicals have good reduction resistance and spin relaxation capabilities. The authors have synthesized these compounds in good yields using conventional, efficient techniques. The dispirocyclic nitroxides studied can lead to different markers and seems promising for these applications. This article is therefore of interest to researchers working in this field and deserves to be published. However, there is room for improvement in the description of the compounds synthesized. The description of the products, particularly in terms of NMR, is not in line with the journal's requirements. I feel that a more succinct description should be included in the article (without attribution), and that the description currently given should be transferred to the supporting information section. This would make reading this section much smoother.

Author Response

Dear Reviewer,

Thank you for your agreement to review the manuscript and for your positive response.

Following your recommendation:

“However, there is room for improvement in the description of the compounds synthesized. The description of the products, particularly in terms of NMR, is not in line with the journal's requirements. I feel that a more succinct description should be included in the article (without attribution), and that the description currently given should be transferred to the supporting information section.”

The detailed discussion of the spectral data was removed from the “results and discussion” section with a reference to SI (see the new version of the manuscript).

We didn’t change spectral description in the “materials and methods” section because it follows the pattern shown in papers of the Special Issue, see, for example https://doi.org/10.3390/ijms241411315 .

We sincerely hope that now the paper looks better.

Igor A. Kirilyuk

Reviewer 2 Report

The group of authors, with valuable experience in the field, addresses an interesting aspect related to the nitroxide class with action of biogenic reductants and high spin relaxation (dephasing) times. The compound (1R(S),5R(S),7R(S),8R(S))-1,8-bis(hydroxymethyl)-6-aza-20 dispiro[4.1.4.2]tridecane-6-oxyl presented in this paper combines these features.

After a comprehensive introduction that refers both to the applications of nitroxides class and to the possibilities of obtaining some individuals from this class, the authors focus on spirocyclic nitroxides. The superiority of the compound presented is demonstrated in comparison to the known similar compound 1, which presents a lower resistance to reduction, obtained by the same authors.

An interesting synthesis is presented in detail, that starts from relatively simple compounds, and step by step leads to well defined and characterized intermediates and products; the yields are from acceptable to very good. The reaction conditions presented on the Scheme help the reader.

The obtaining of radical 22 and the study of its transformation into the mixture of isomers 23 seems interesting. The two hydroxymethyl groups near the radical center, give the possibility of synthesizing mono- and bifunctional spin labels. Therefore, next, several alkylation and acylation reactions of compound 5 towards functional derivatives were carried out. Finally, the spin labels capable of binding azides, the nitroxides 30 and 31, were prepared via alkylation of 5 with propargyl bromide.The ability of the spin label 31 to bind to azide-containing biomolecules was demonstrated by the reaction with 2,3,4,6-tetra-O-acetyl-β-D-galactopyranosyl azide. The results of reduction kinetics of nitroxides 5, 29 and 33 with ascorbate were also presented.

The experimental part is correctly presented and all the appropriate procedures were used to assign the structure of intermediates and products. The contribution of single-crystal X-ray diffraction data to establishing the structure of the compounds is noteworthy.

In conclusion, the article can be published in the present form.

Author Response

Dear Reviewer,

Thank you for your agreement to review the manuscript and for your positive response. 

Igor A. Kirilyuk

Reviewer 3 Report

Dear Editor,

I would like to thank you for inviting me to review the paper. The study is interesting and can benefit the scientific public. The authors have processed their study in detail, the results are clearly presented. Based on all of the above, I suggest that the study be accepted with minor corrections. The study is also acceptable in its original form. Below you can find my comment.

The Introduction part is written too generally. It should be rearranged so that it better suits the topic that the authors deal with, with reference to some of the most important references and their results.

Author Response

Dear Reviewer,

Thank you for your agreement to review the manuscript and for your positive response.

You wrote:

“The Introduction part is written too generally. It should be rearranged so that it better suits the topic that the authors deal with, with reference to some of the most important references and their results.”

In this manuscript we tried to follow Instructions for Authors where it is indicated that “The introduction should briefly place the study in a broad context and highlight why it is important. It should define the purpose of the work and its significance, including specific hypotheses being tested. The current state of the research field should be reviewed carefully and key publications cited.”, and “Finally, briefly mention the main aim of the work and highlight the main conclusions. Keep the introduction comprehensible to scientists working outside the topic of the paper.”

Indeed, the relevance and importance of the study is shown in the paragraph 1.

The current state of the research is given in the paragraph 2 and 4.

Specific hypothesis is formulated in the paragraph 3.

The main results and conclusions are summarized in the paragraph 5.

Do you really think that these paragraphs does not suit well to the topic we deal with?

Sorry, but I do not understand, what did we miss? Are there any relevant references or important results which you want us to include?

Sincerely,

Igor A. Kirilyuk